# DIVERSITY IS ALL YOU NEED:
# LEARNING SKILLS WITHOUT A REWARD FUNCTION

**Benjamin Eysenbach** *
Carnegie Mellon University, Google Brain
beysenba@cs.cmu.edu

**Abhishek Gupta**
UC Berkeley
abhigupta@berkeley.edu

**Julian Ibarz**
Google Brain
julianibarz@google.com

**Sergey Levine**
UC Berkeley, Google Brain
svlevine@eecs.berkeley.edu

## ABSTRACT

Intelligent creatures can explore their environments and learn useful skills without supervision. In this paper, we propose "Diversity is All You Need"(DIAYN), a method for learning useful skills without a reward function. Our proposed method learns skills by maximizing an information theoretic objective using a maximum entropy policy. On a variety of simulated robotic tasks, we show that this simple objective results in the unsupervised emergence of diverse skills, such as walking and jumping. In a number of reinforcement learning benchmark environments, our method is able to learn a skill that solves the benchmark task despite never receiving the true task reward. We show how pretrained skills can provide a good parameter initialization for downstream tasks, and can be composed hierarchically to solve complex, sparse reward tasks. Our results suggest that unsupervised discovery of skills can serve as an effective pretraining mechanism for overcoming challenges of exploration and data efficiency in reinforcement learning.

## 1 INTRODUCTION

Deep reinforcement learning (RL) has been demonstrated to effectively learn a wide range of reward-driven skills, including playing games (Mnih et al., 2013; Silver et al., 2016), controlling robots (Gu et al., 2017; Schulman et al., 2015b), and navigating complex environments (Zhu et al., 2017; Mirowski et al., 2016). However, intelligent creatures can explore their environments and learn useful skills even without supervision, so that when they are later faced with specific goals, they can use those skills to satisfy the new goals quickly and efficiently.

Learning skills without reward has several practical applications. Environments with sparse rewards effectively have no reward until the agent randomly reaches a goal state. Learning useful skills without supervision may help address challenges in exploration in these environments. For long horizon tasks, skills discovered without reward can serve as primitives for hierarchical RL, effectively shortening the episode length. In many practical settings, interacting with the environment is essentially free, but evaluating the reward requires human feedback (Christiano et al., 2017). Unsupervised learning of skills may reduce the amount of supervision necessary to learn a task. While we can take the human out of the loop by designing a reward function, it is challenging to design a reward function that elicits the desired behaviors from the agent (Hadfield-Menell et al., 2017). Finally, when given an unfamiliar environment, it is challenging to determine what tasks an agent should be able to learn. Unsupervised skill discovery partially answers this question.[1]

Autonomous acquisition of useful skills without any reward signal is an exceedingly challenging problem. A *skill* is a latent-conditioned policy that alters the state of the environment in a consistent way. We consider the setting where the reward function is unknown, so we want to learn a set of skills by maximizing the utility of this set. Making progress on this problem requires specifying a

---

*Work done as a member of the Google AI Residency Program (g.co/airesidency).
[1]See videos here: https://sites.google.com/view/diayn/

learning objective that ensures that each skill individually is distinct and that the skills collectively explore large parts of the state space. In this paper, we show how a simple objective based on mutual information can enable RL agents to autonomously discover such skills. These skills are useful for a number of applications, including hierarchical reinforcement learning and imitation learning.

We propose a method for learning diverse skills with deep RL in the absence of any rewards. We hypothesize that in order to acquire skills that are useful, we must train the skills so that they maximize coverage over the set of possible behaviors. While one skill might perform a useless behavior like random dithering, other skills should perform behaviors that are distinguishable from random dithering, and therefore more useful. A key idea in our work is to use discriminability between skills as an objective. Further, skills that are distinguishable are not necessarily maximally diverse – a slight difference in states makes two skills distinguishable, but not necessarily diverse in a semantically meaningful way. To combat this problem, we want to learn skills that not only are distinguishable, but also are *as diverse as possible*. By learning distinguishable skills that are as random as possible, we can "push" the skills away from each other, making each skill robust to perturbations and effectively exploring the environment. By maximizing this objective, we can learn skills that run forward, do backflips, skip backwards, and perform face flops (see Figure 3).

Our paper makes five contributions. First, we propose a method for learning useful skills without any rewards. We formalize our discriminability goal as maximizing an information theoretic objective with a maximum entropy policy. Second, we show that this simple exploration objective results in the unsupervised emergence of diverse skills, such as running and jumping, on several simulated robotic tasks. In a number of RL benchmark environments, our method is able to solve the benchmark task despite never receiving the true task reward. In these environments, some of the learned skills correspond to solving the task, and each skill that solves the task does so in a distinct manner. Third, we propose a simple method for using learned skills for hierarchical RL and find this methods solves challenging tasks. Four, we demonstrate how skills discovered can be quickly adapted to solve a new task. Finally, we show how skills discovered can be used for imitation learning.

## 2 RELATED WORK

Previous work on hierarchical RL has learned skills to maximize a single, known, reward function by jointly learning a set of skills and a meta-controller (e.g., (Bacon et al., 2017; Heess et al., 2016; Dayan & Hinton, 1993; Frans et al., 2017; Krishnan et al., 2017; Florensa et al., 2017)). One problem with joint training (also noted by Shazeer et al. (2017)) is that the meta-policy does not select "bad" options, so these options do not receive any reward signal to improve. Our work prevents this degeneracy by using a random meta-policy during unsupervised skill-learning, such that neither the skills nor the meta-policy are aiming to solve any single task. A second importance difference is that our approach learns skills *with no reward*. Eschewing a reward function not only avoids the difficult problem of reward design, but also allows our method to learn task-agnostic.

Related work has also examined connections between RL and information theory (Ziebart et al., 2008; Schulman et al., 2017; Nachum et al., 2017; Haarnoja et al., 2017) and developed maximum entropy algorithms with these ideas Haarnoja et al. (2018; 2017). Recent work has also applied tools from information theory to skill discovery. Mohamed & Rezende (2015) and Jung et al. (2011) use the mutual information between states and actions as a notion of empowerment for an intrinsically motivated agent. Our method maximizes the mutual information between states and *skills*, which can be interpreted as maximizing the empowerment of a *hierarchical agent* whose action space is the set of skills. Hausman et al. (2018), Florensa et al. (2017), and Gregor et al. (2016) showed that a discriminability objective is equivalent to maximizing the mutual information between the latent skill $z$ and some aspect of the corresponding trajectory. Hausman et al. (2018) considered the setting with many tasks and reward functions and Florensa et al. (2017) considered the setting with a single task reward. Three important distinctions allow us to apply our method to tasks significantly more complex than the gridworlds in Gregor et al. (2016). First, we use maximum entropy policies to force our skills to be diverse. Our theoretical analysis shows that including entropy maximization in the RL objective results in the mixture of skills being maximum entropy in aggregate. Second, we fix the prior distribution over skills, rather than learning it. Doing so prevents our method from collapsing to sampling only a handful of skills. Third, while the discriminator in Gregor et al. (2016) only looks at

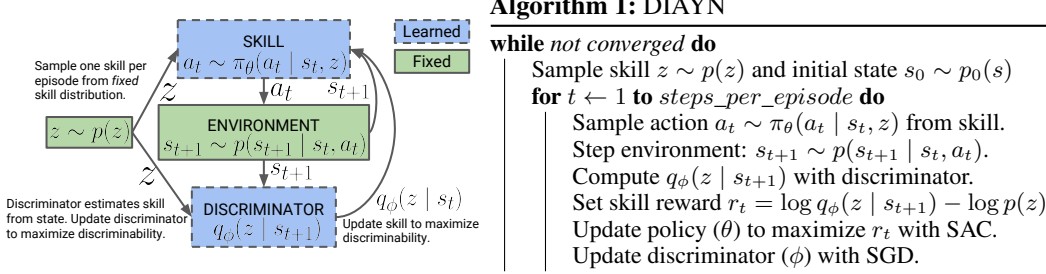

Figure 1: **DIAYN Algorithm**: We update the discriminator to better predict the skill, and update the skill to visit diverse states that make it more discriminable.

the final state, our discriminator looks at every state, which provides additional reward signal. These three crucial differences help explain how our method learns useful skills in complex environments.

Prior work in neuroevolution and evolutionary algorithms has studied how complex behaviors can be learned by directly maximizing diversity (Lehman & Stanley, 2011a;b; Woolley & Stanley, 2011; Stanley & Miikkulainen, 2002; Such et al., 2017; Pugh et al., 2016; Mouret & Doncieux, 2009). While this prior work uses diversity maximization to obtain better solutions, we aim to acquire complex skills with minimal supervision to improve efficiency (i.e., reduce the number of objective function queries) and as a stepping stone for imitation learning and hierarchical RL. We focus on deriving a general, information-theoretic objective that does not require manual design of distance metrics and can be applied to any RL task without additional engineering.

Previous work has studied intrinsic motivation in humans and learned agents. Ryan & Deci (2000); Bellemare et al. (2016); Fu et al. (2017); Schmidhuber (2010); Oudeyer et al. (2007); Pathak et al. (2017); Baranes & Oudeyer (2013). While these previous works use an intrinsic motivation objective to learn a *single* policy, we propose an objective for learning *many*, diverse policies. Concurrent work Achiam et al. (2017) draws ties between learning discriminable skills and variational autoencoders. We show that our method scales to more complex tasks, likely because of algorithmic design choices, such as our use of an off-policy RL algorithm and conditioning the discriminator on individual states.

## 3 DIVERSITY IS ALL YOU NEED

We consider an unsupervised RL paradigm in this work, where the agent is allowed an unsupervised "exploration" stage followed by a supervised stage. In our work, the aim of the unsupervised stage is to learn skills that eventually will make it easier to maximize the task reward in the supervised stage. Conveniently, because skills are learned without a priori knowledge of the task, the learned skills can be used for many different tasks.

### 3.1 HOW IT WORKS

Our method for unsupervised skill discovery, DIAYN ("Diversity is All You Need"), builds off of three ideas. First, for skills to be useful, we want the skill to dictate the states that the agent visits. Different skills should visit different states, and hence be distinguishable. Second, we want to use states, not actions, to distinguish skills, because actions that do not affect the environment are not visible to an outside observer. For example, an outside observer cannot tell how much force a robotic arm applies when grasping a cup if the cup does not move. Finally, we encourage exploration and incentivize the skills to be as diverse as possible by learning skills that act as randomly as possible. Skills with high entropy that remain discriminable must explore a part of the state space far away from other skills, lest the randomness in its actions lead it to states where it cannot be distinguished.

We construct our objective using notation from information theory: $S$ and $A$ are random variables for states and actions, respectively; $Z \sim p(z)$ is a latent variable, on which we condition our policy; we refer to a the policy conditioned on a fixed $Z$ as a "skill"; $I(\cdot; \cdot)$ and $\mathcal{H}[\cdot]$ refer to mutual

information and Shannon entropy, both computed with base $e$. In our objective, we maximize the mutual information between skills and states, $I(S; Z)$, to encode the idea that the skill should control which states the agent visits. Conveniently, this mutual information dictates that we can infer the skill from the states visited. To ensure that states, not actions, are used to distinguish skills, we minimize the mutual information between skills and actions given the state, $I(A; Z \mid S)$. Viewing all skills together with $p(z)$ as a mixture of policies, we maximize the entropy $\mathcal{H}[A \mid S]$ of this mixture policy. In summary, we maximize the following objective with respect to our policy parameters, $\theta$:

$$\mathcal{F}(\theta) \triangleq I(S; Z) + \mathcal{H}[A \mid S] - I(A; Z \mid S) \tag{1}$$
$$= (\mathcal{H}[Z] - \mathcal{H}[Z \mid S]) + \mathcal{H}[A \mid S] - (\mathcal{H}[A \mid S] - \mathcal{H}[A \mid S, Z])$$
$$= \mathcal{H}[Z] - \mathcal{H}[Z \mid S] + \mathcal{H}[A \mid S, Z] \tag{2}$$

We rearranged our objective in Equation 2 to give intuition on how we optimize it.[2] The first term encourages our prior distribution over $p(z)$ to have high entropy. We fix $p(z)$ to be uniform in our approach, guaranteeing that it has maximum entropy. The second term suggests that it should be easy to infer the skill $z$ from the current state. The third term suggests that each skill should act as randomly as possible, which we achieve by using a maximum entropy policy to represent each skill. As we cannot integrate over all states and skills to compute $p(z \mid s)$ exactly, we approximate this posterior with a learned discriminator $q_\phi(z \mid s)$. Jensen's Inequality tells us that replacing $p(z \mid s)$ with $q_\phi(z \mid s)$ gives us a variational lower bound $\mathcal{G}(\theta, \phi)$ on our objective $\mathcal{F}(\theta)$ (see (Agakov, 2004) for a detailed derivation):

$$\mathcal{F}(\theta) = \mathcal{H}[A \mid S, Z] - \mathcal{H}[Z \mid S] + \mathcal{H}[Z]$$
$$= \mathcal{H}[A \mid S, Z] + \mathbb{E}_{z \sim p(z), s \sim \pi(z)}[\log p(z \mid s)] - \mathbb{E}_{z \sim p(z)}[\log p(z)]$$
$$\geq \mathcal{H}[A \mid S, Z] + \mathbb{E}_{z \sim p(z), s \sim \pi(z)}[\log q_\phi(z \mid s) - \log p(z)] \triangleq \mathcal{G}(\theta, \phi)$$

## 3.2 IMPLEMENTATION

We implement DIAYN with soft actor critic (SAC) (Haarnoja et al., 2018), learning a policy $\pi_\theta(a \mid s, z)$ that is conditioned on the latent variable $z$. Soft actor critic maximizes the policy's entropy over actions, which takes care of the entropy term in our objective $\mathcal{G}$. Following Haarnoja et al. (2018), we scale the entropy regularizer $\mathcal{H}[a \mid s, z]$ by $\alpha$. We found empirically that an $\alpha = 0.1$ provided a good trade-off between exploration and discriminability. We maximize the expectation in $\mathcal{G}$ by replacing the task reward with the following pseudo-reward:

$$r_z(s, a) \triangleq \log q_\phi(z \mid s) - \log p(z) \tag{3}$$

We use a categorical distribution for $p(z)$. During unsupervised learning, we sample a skill $z \sim p(z)$ at the start of each episode, and act according to that skill throughout the episode. The agent is rewarded for visiting states that are easy to discriminate, while the discriminator is updated to better infer the skill $z$ from states visited. Entropy regularization occurs as part of the SAC update.

## 3.3 STABILITY

Unlike prior adversarial unsupervised RL methods (e.g., Sukhbaatar et al. (2017)), DIAYN forms a cooperative game, which avoids many of the instabilities of adversarial saddle-point formulations. On gridworlds, we can compute analytically that the unique optimum to the DIAYN optimization problem is to evenly partition the states between skills, with each skill assuming a uniform stationary distribution over its partition (proof in Appendix B). In the continuous and approximate setting, convergence guarantees would be desirable, but this is a very tall order: even standard RL methods with function approximation (e.g., DQN) lack convergence guarantees, yet such techniques are still useful. Empirically, we find DIAYN to be robust to random seed; varying the random seed does not noticeably affect the skills learned, and has little effect on downstream tasks (see Fig.s 4, 6, and 13).

## 4 EXPERIMENTS

In this section, we evaluate DIAYN and compare to prior work. First, we analyze the skills themselves, providing intuition for the types of skills learned, the training dynamics, and how we avoid problematic

---

[2]While our method uses stochastic policies, note that for deterministic policies in continuous action spaces, $I(A; Z \mid S) = \mathcal{H}[A \mid S]$. Thus, for deterministic policies, Equation 2 reduces to maximizing $I(S; Z)$.

behavior in previous work. In the second half, we show how the skills can be used for downstream tasks, via policy initialization, hierarchy, imitation, outperforming competitive baselines on most tasks. We encourage readers to view videos[3] and code[4] for our experiments.

## 4.1 ANALYSIS OF LEARNED SKILLS

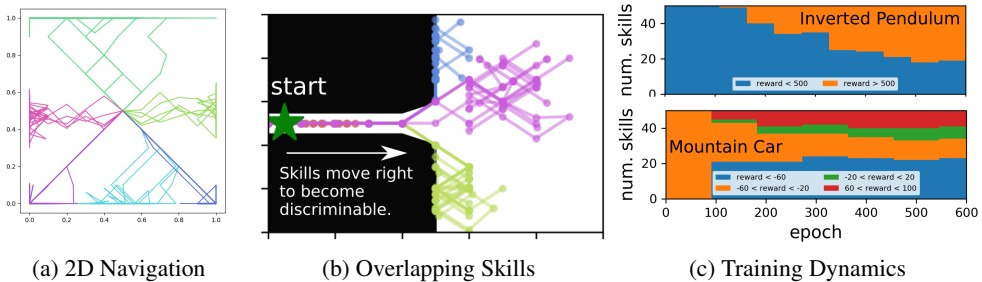

(a) 2D Navigation    (b) Overlapping Skills    (c) Training Dynamics

Figure 2: *(Left)* DIAYN skills in a simple navigation environment; *(Center)* skills can overlap if they eventually become distinguishable; *(Right)* diversity of the rewards increases throughout training.

**Question 1.** *What skills does DIAYN learn?*

We study the skills learned by DIAYN on tasks of increasing complexity, ranging from point navigation (2 dimensions) to ant locomotion (111 dimensions). We first applied DIAYN to a simple 2D navigation environment. The agent starts in the center of the box, and can take actions to directly move its $(x, y)$ position. Figure 2a illustrates how the 6 skills learned for this task move away from each other to remain distinguishable. Next, we applied DIAYN to two classic control tasks, inverted pendulum and mountain car. Not only does our approach learn skills that solve the task without rewards, it learns multiple distinct skills for solving the task. (See Appendix D for further analysis.)

Finally, we applied DIAYN to three continuous control tasks (Brockman et al., 2016): half cheetah, hopper, and ant. As shown in Figure 3, we learn a diverse set of primitive behaviors for all tasks. For half cheetah, we learn skills for running forwards and backwards at various speeds, as well as skills for doing flips and falling over; ant learns skills for jumping and walking in many types of curved trajectories (though none walk in a straight line); hopper learns skills for balancing, hopping forward and backwards, and diving. See Appendix D.4 for a comparison with VIME.

**Question 2.** *How does the distribution of skills change during training?*

While DIAYN learns skills without a reward function, as an outside observer, can we evaluate the skills throughout training to understand the training dynamics. Figure 2 shows how the skills for inverted pendulum and mountain car become increasingly diverse throughout training (Fig. 13 repeats this experiment for 5 random seeds, and shows that results are robust to initialization). Recall that our skills are learned with no reward, so it is natural that some skills correspond to small task reward while others correspond to large task reward.

**Question 3.** *Does discriminating on single states restrict DIAYN to learn skills that visit disjoint sets of states?*

Our discriminator operates at the level of states, not trajectories. While DIAYN favors skills that do not overlap, our method is not limited to learning skills that visit entirely disjoint sets of states. Figure 2b shows a simple experiment illustrating this. The agent starts in a hallway (green star), and can move more freely once exiting the end of the hallway into a large room. Because RL agents are incentivized to maximize their cumulative reward, they may take actions that initially give no reward to reach states that eventually give high reward. In this environment, DIAYN learns skills that exit the hallway to make them mutually distinguishable.

---

[3]https://sites.google.com/view/diayn/
[4]https://github.com/ben-eysenbach/sac/blob/master/DIAYN.md

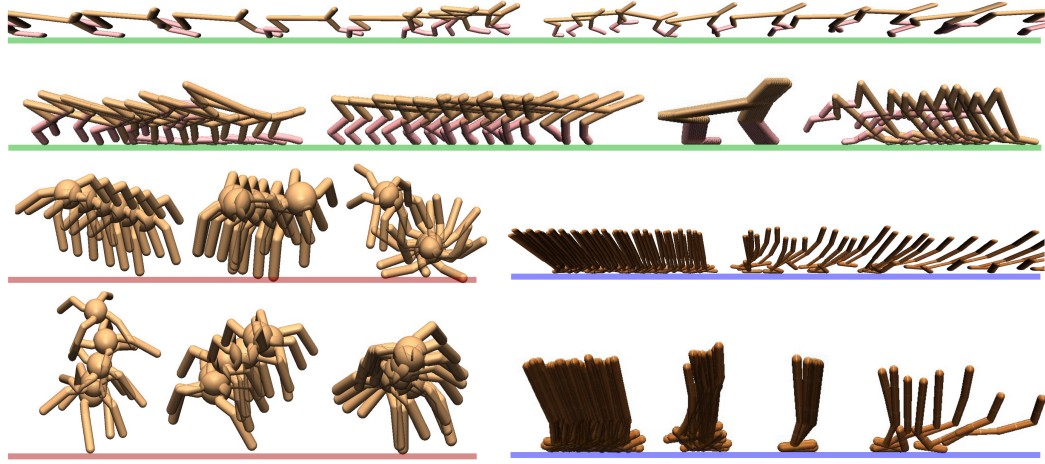

Figure 3: **Locomotion skills**: Without any reward, DIAYN discovers skills for running, walking, hopping, flipping, and gliding. It is challenging to craft reward functions that elicit these behaviors.

**Question 4.** *How does DIAYN differ from Variational Intrinsic Control (VIC) (Gregor et al., 2016)?*

The key difference from the most similar prior work on unsupervised skill discovery, VIC, is our decision to *not* learn the prior $p(z)$. We found that VIC suffers from the "Matthew Effect" Merton (1968): VIC's learned prior $p(z)$ will sample the more diverse skills more frequently, and hence only those skills will receive training signal to improve. To study this, we evaluated DIAYN and VIC on the half-cheetah environment, and plotting the effective number of skills (measured as $\exp(\mathcal{H}[Z])$) throughout

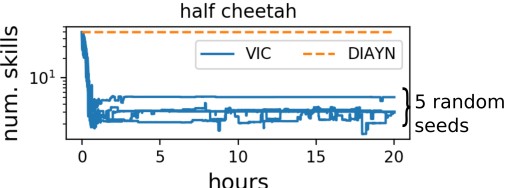

Figure 4: **Why use a fixed prior?** In contrast to prior work, DIAYN continues to sample all skills throughout training.

training (details and more figures in Appendix E.2). The figure to the right shows how VIC quickly converges to a setting where it only samples a handful of skills. In contrast, DIAYN fixes the distribution over skills, which allows us to discover more diverse skills.

## 4.2 HARNESSING LEARNED SKILLS

The perhaps surprising finding that we can discover diverse skills without a reward function creates a building block for many problems in RL. For example, to find a policy that achieves a high reward on a task, it is often sufficient to simply choose the skill with largest reward. Three less obvious applications are adapting skills to maximize a reward, hierarchical RL, and imitation learning.

### 4.2.1 ACCELERATING LEARNING WITH POLICY INITIALIZATION

After DIAYN learns task-agnostic skills without supervision, we can quickly adapt the skills to solve a desired task. Akin to the use of pre-trained models in computer vision, we propose that DIAYN can serve as unsupervised pre-training for more sample-efficient finetuning of task-specific policies.

**Question 5.** *Can we use learned skills to directly maximize the task reward?*

We take the skill with highest reward for each benchmark task and further finetune this skill using the task-specific reward function. We compare to a "random initialization" baseline that is initialized from scratch. Our approach differs from this baseline only in how weights are initialized. We initialize both the policy and value networks with weights learned during unsupervised pretraining. Although the critic networks learned during pretraining corresponds to the pseudo-reward from the discriminator (Eq. 3) and not the true task reward, we found empirically that the pseudo-reward was close to the true task reward for the best skill, and initializing the critic in addition to the actor further sped up learning. Figure 5 shows both methods applied to half cheetah, hopper, and ant. We assume

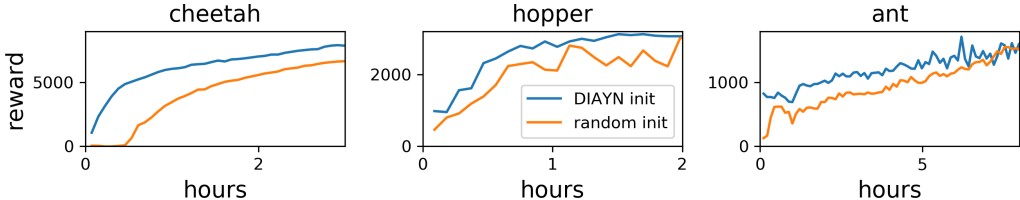

Figure 5: **Policy Initialization**: Using a DIAYN skill to initialize weights in a policy accelerates learning, suggesting that pretraining with DIAYN may be especially useful in resource constrained settings. Results are averages across 5 random seeds.

that the unsupervised pretraining is free (e.g., only the reward function is expensive to compute) or can be amortized across many tasks, so we omit pretraining steps from this plot. On all tasks, unsupervised pretraining enables the agent to learn the benchmark task more quickly.

#### 4.2.2 USING SKILLS FOR HIERARCHICAL RL

In theory, hierarchical RL should decompose a complex task into motion primitives, which may be reused for multiple tasks. In practice, algorithms for hierarchical RL can encounter many problems: (1) each motion primitive reduces to a single action (Bacon et al., 2017), (2) the hierarchical policy only samples a single motion primitive (Gregor et al., 2016), or (3) all motion primitives attempt to do the entire task. In contrast, DIAYN discovers diverse, *task-agnostic* skills, which hold the promise of acting as a building block for hierarchical RL.

**Question 6.** *Are skills discovered by DIAYN useful for hierarchical RL?*

We propose a simple extension to DIAYN for hierarchical RL, and find that simple algorithm outperforms competitive baselines on two challenging tasks. To use the discovered skills for hierarchical RL, we learn a meta-controller whose actions are to choose which skill to execute for the next $k$ steps (100 for ant navigation, 10 for cheetah hurdle). The meta-controller has the same observation space as the skills.

As an initial test, we applied the hierarchical RL algorithm to a simple 2D point navigation task (details in Appendix C.2). Figure 6 illustrates how the reward on this task increases with the number of skills; error bars show the standard deviation across 5 random seeds. To ensure that our goals were not cherry picked, we sampled 25 goals evenly from the state space, and evaluated each random seed on all goals. We also compared to Variational Information Maximizing Exploration

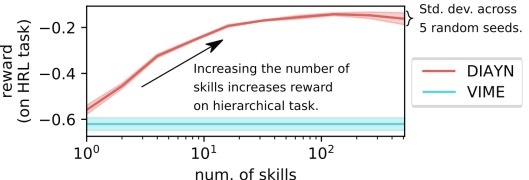

Figure 6: **Hierarchical RL**

(VIME) (Houthooft et al., 2016). Note that even the best random seed from VIME significantly under-performs DIAYN. This is not surprising: whereas DIAYN learns a set of skills that effectively partition the state space, VIME attempts to learn a single policy that visits many states.

Next, we applied the hierarchical algorithm to two challenging simulated robotics environment. On the cheetah hurdle task, the agent is rewarded for bounding up and over hurdles, while in the ant navigation task, the agent must walk to a set of 5 waypoints in a specific order, receiving only a sparse reward upon reaching each waypoint. The sparse reward and obstacles in these environments make them exceedingly difficult for

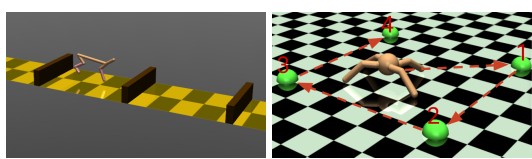

Figure 7: Challenging tasks for hierarchical RL: *(Left)* Cheetah Hurdle; *(Right)* Ant Navigation

non-hierarchical RL algorithms. Indeed, state of the art RL algorithms that do not use hierarchies perform poorly on these tasks. Figure 8 shows how DIAYN outperforms state of the art on-policy RL (TRPO (Schulman et al., 2015a)), off-policy RL (SAC (Haarnoja et al., 2018)), and exploration bonuses (VIME). This experiment suggests that unsupervised skill learning provides an effective mechanism for combating challenges of exploration and sparse rewards in RL.

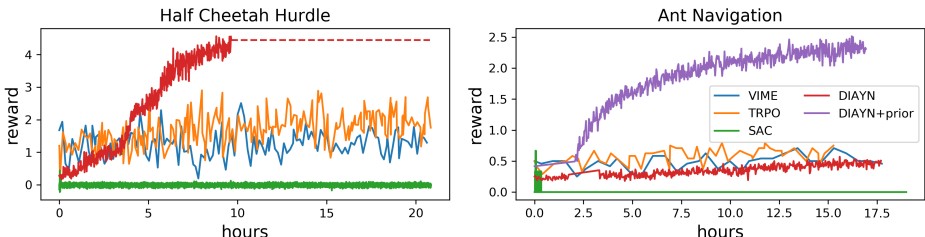

Figure 8: **DIAYN for Hierarchical RL**: By learning a meta-controller to compose skills learned by DIAYN, cheetah quickly learns to jump over hurdles and ant solves a sparse-reward navigation task.

**Question 7.** *How can DIAYN leverage prior knowledge about what skills will be useful?*

If the number of possible skills grows exponentially with the dimension of the task observation, one might imagine that DIAYN would fail to learn skills necessary to solve some tasks. While we found that DIAYN *does* scale to tasks with more than 100 dimensions (ant has 111), we can also use a simple modification to bias DIAYN towards discovering particular types of skills. We can condition the discriminator on only a subset of the observation space, or any other function of the observations. In this case, the discriminator maximizes $\mathbb{E}[\log q_\phi(z \mid f(s))]$. For example, in the ant navigation task, $f(s)$ could compute the agent's center of mass, and DIAYN would learn skills that correspond to changing the center of mass. The "DIAYN+prior" result in Figure 8 (right) shows how incorporating this prior knowledge can aid DIAYN in discovering useful skills and boost performance on the hierarchical task. (No other experiments or figures in this paper used this prior.) The key takeaway is that while DIAYN is primarily an unsupervised RL algorithm, there is a simple mechanism for incorporating supervision when it is available. Unsurprisingly, we perform better on hierarchical tasks when incorporating more supervision.

### 4.2.3 IMITATING AN EXPERT

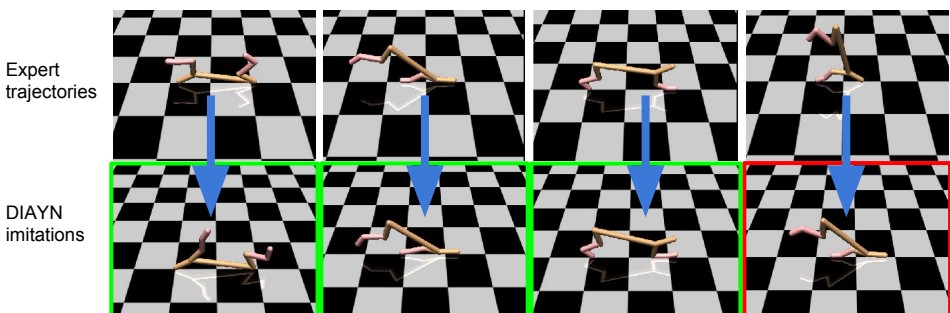

Figure 9: **Imitating an expert**: DIAYN imitates an expert standing upright, flipping, and faceplanting, but fails to imitate a handstand.

**Question 8.** *Can we use learned skills to imitate an expert?*

Aside from maximizing reward with finetuning and hierarchical RL, we can also use learned skills to follow expert demonstrations. One use-case is where a human manually controls the agent to complete a task that we would like to automate. Simply replaying the human's actions fails in stochastic environments, cases where closed-loop control is necessary. A second use-case involves an existing agent with a hard coded, manually designed policy. Imitation learning replaces the existing policy with a similar yet differentiable policy, which might be easier to update in response to new constraints or objectives. We consider the setting where we are given an expert trajectory consisting of states, without actions, defined as $\tau^* = \{(s_i)\}_{1 \leq i \leq N}$. Our goal is to obtain a feedback controller that will reach the same states. Given the expert trajectory, we use our learned discriminator to estimate which skill was most likely to have generated the trajectory. This optimization problem,

which we solve for categorical $z$ by enumeration, is equivalent to an M-projection (Bishop, 2016):

$$\hat{z} = \arg\max_{z} \Pi_{s_t \in \tau^*} q_\phi(z \mid s_t)$$

We qualitatively evaluate this approach to imitation learning on half cheetah. Figure 9 (left) shows four imitation tasks, three of which our method successfully imitates. We quantitatively evaluate this imitation method on classic control tasks in Appendix G.

## 5 CONCLUSION

In this paper, we present DIAYN, a method for learning skills without reward functions. We show that DIAYN learns diverse skills for complex tasks, often solving benchmark tasks with one of the learned skills without actually receiving any task reward. We further proposed methods for using the learned skills (1) to quickly adapt to a new task, (2) to solve complex tasks via hierarchical RL, and (3) to imitate an expert. As a rule of thumb, DIAYN may make learning a task easier by replacing the task's complex action space with a set of useful skills. DIAYN could be combined with methods for augmenting the observation space and reward function. Using the common language of information theory, a joint objective can likely be derived. DIAYN may also more efficiently learn from human preferences by having humans select among learned skills. Finally, the skills produced by DIAYN might be used by game designers to allow players to control complex robots and by artists to animate characters.

**Acknowledgements:** We'd like to thank JD Co-Reyes and Andrew Liu for insightful discussions, and our anonymous reviewers for their thoughtful feedback and suggestions.

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

## A  PSEUDO-REWARD

The $\log p(z)$ term in Equation 3 is a baseline that does not depend on the policy parameters $\theta$, so one might be tempted to remove it from the objective. We provide a two justifications for keeping it. First, assume that episodes never terminate, but all skills eventually converge to some absorbing state (e.g., with all sensors broken). At this state, the discriminator cannot distinguish the skills, so its estimate is $\log q(z \mid s) = \log(1/N)$, where $N$ is the number of skills. For practical reasons, we want to restart the episode after the agent reaches the absorbing state. Subtracting $\log(z)$ from the pseudo-reward at every time step in our finite length episodes is equivalent to pretending that episodes never terminate and the agent gets reward $\log(z)$ after our "artificial" termination. Second, assuming our discriminator $q_\phi$ is better than chance, we see that $q_\phi(z \mid s) \geq p(z)$. Thus, subtracting the $\log p(z)$ baseline ensures our reward function is always non-negative, encouraging the agent to stay alive. Without this baseline, an optimal agent would end the episode as soon as possible.[5]

## B  OPTIMUM FOR GRIDWORLDS

For simple environments, we can compute an analytic solution to the DIAYN objective. For example, consider a $N \times N$ gridworld, where actions are to move up/down/left/right. Any action can be taken in any state, but the agent will stay in place if it attempts to move out of the gridworld. We use $(x, y)$ to refer to states, where $x, y \in \{1, 2, \cdots, N\}$.

For simplicity, we assume that, for every skill, the distribution of states visited exactly equals that skill's stationary distribution over states. To clarify, we will use $\pi_z$ to refer to the policy for skill $z$. We use $\rho_{\pi_z}$ to indicate skill $z$'s stationary distribution over states, and $\hat{\rho}_{\pi_z}$ as the empirical distribution over states within a single episode. Our assumption is equivalent to saying

$$\rho_{\pi_z}(s) = \hat{\rho}_{\pi_z}(s) \qquad \forall s \in \mathcal{S}$$

One way to ensure this is to assume infinite-length episodes.

We want to show that a set of skills that evenly partitions the state space is the optimum of the DIAYN objective for this task. While we will show this only for the 2-skill case, the 4 skill case is analogous.

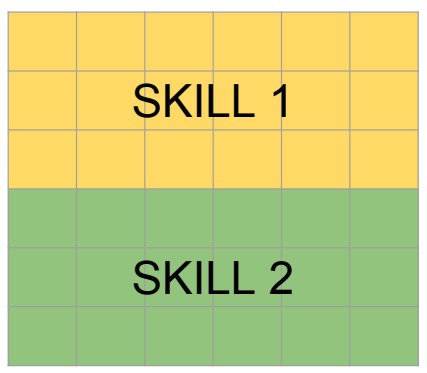

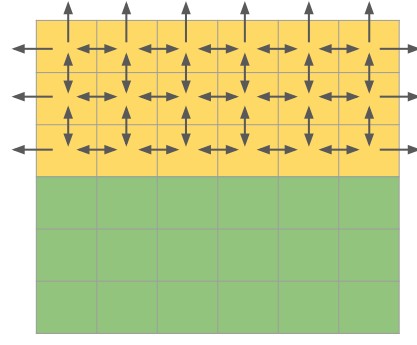

(b) Policy for one of the optimal skills. The agent stays
(a) Optimum Skills for Gridworld with 2 Skills  in place when it attempts to leave the gridworld.

Figure 10: **Optimum for Gridworlds:** For gridworld environments, we can compute an analytic solution to the DIAYN objective.

The optimum policies for a set of two skills are those which evenly partition the state space. We will show that a top/bottom partition is one such (global) optima. The left/right case is analogous.

**Lemma B.1.** *A pair of skills with state distributions given below (and shown in Figure 10) are an optimum for the DIAYN objective with no entropy regularization ($\alpha = 0$).*

$$\rho_{\pi_1}(x, y) = \frac{2}{N^2}\delta(y \leq N/2) \quad and \quad \rho_{\pi_2}(x, y) = \frac{2}{N^2}\delta(y > N/2) \tag{4}$$

---

[5]In some environments, such as mountain car, it is desirable for the agent to end the episode as quickly as possible. For these types of environments, the $\log p(z)$ baseline can be removed.

Before proving Lemma B.1, we note that there exist policies that achieve these stationary distributions. Figure 10b shows one such policy, were each arrow indicates a transition with probability $\frac{1}{4}$. Note that when the agent is in the bottom row of yellow states, it does not transition to the green states, and instead stays in place with probability $\frac{1}{4}$. Note that the distribution in Equation 4 satisfies the detailed balance equations (Murphy, 2012).

*Proof.* Recall that the DIAYN objective with no entropy regularization is:

$$-\mathcal{H}[Z \mid S] + \mathcal{H}[Z]$$

Because the skills partition the states, we can always infer the skill from the state, so $\mathcal{H}[Z \mid S] = 0$. By construction, the prior distribution over $\mathcal{H}[Z]$ is uniform, so $\mathcal{H}[Z] = \log(2)$ is maximized. Thus, a set of two skills that partition the state space maximizes the un-regularized DIAYN objective. □

Next, we consider the regularized objective. In this case, we will show that while an even partition is not perfectly optimal, it is "close" to optimal, and its "distance" from optimal goes to zero as the gridworld grows in size. This analysis will give us additional insight into the skills preferred by the DIAYN objective.

**Lemma B.2.** *A pair of skills with state distributions given given in Equation 4 achieve an DIAYN objective within a factor of $O(1/N)$ of the optimum, where $N$ is the gridworld size.*

*Proof.* Recall that the DIAYN objective with no entropy regularization is:

$$\mathcal{H}[A \mid S, Z] - \mathcal{H}[Z \mid S] + \mathcal{H}[Z]$$

We have already computed the second two terms in the previous proof: $\mathcal{H}[Z \mid S] = 0$ and $\mathcal{H}[Z] = \log(2)$. For computing the first term, it is helpful to define the set of "border states" for a particular skill as those that do not neighbor another skill. For skill 1 defined in Figure 10 (colored yellow), the border states are: $\{(x, y) \mid y = 4\}$. Now, computing the first term is straightforward:

$$\mathcal{H}[A \mid S, Z] = \frac{2}{N^2} \left( \underbrace{(N/2 - 1)N}_{\text{non-border states}} \log(4) + \underbrace{N}_{\text{border states}} \frac{3}{4} \log(4) \right)$$

$$= \frac{2\log(4)}{N^2} \left( \frac{1}{2}N^2 - \frac{1}{4}N \right)$$

$$= \log(4)(1 - \frac{1}{2N})$$

Thus, the overall objective is within $\frac{\log(4)}{2N}$ of optimum. □

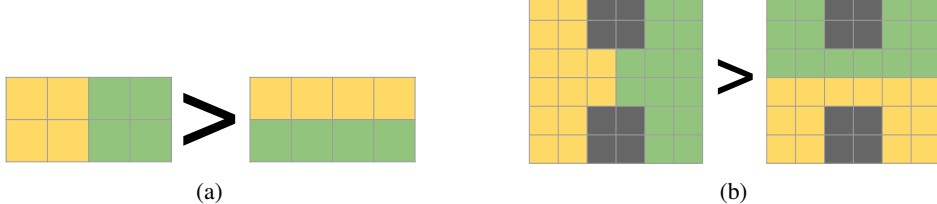

(a)                                   (b)

Figure 11: The DIAYN objective prefers skills that *(Left)* partition states into sets with short borders and *(Right)* which correspond to bottleneck states.

Note that the term for maximum entropy over actions ($\mathcal{H}[A \mid S, Z]$) comes into conflict with the term for discriminability ($-\mathcal{H}[Z \mid S]$) at states along the border between two skills. Everything else being equal, this conflict encourages DIAYN to produce skills that have small borders, as shown in Figure 11. For example, in a gridworld with dimensions $N < M$, a pair of skills that split along the first dimension (producing partitions of size $(N, M/2)$) would achieve a larger (better) objective than skills that split along the second dimension. This same intuition that DIAYN seeks to minimize the border length between skills results in DIAYN preferring partitions that correspond to bottleneck states (see Figure 11b).

## C   Experimental Details

In our experiments, we use the same hyperparameters as those in Haarnoja et al. (2018), with one notable exception. For the Q function, value function, and policy, we use neural networks with 300 hidden units instead of 128 units. We found that increasing the model capacity was necessary to learn many diverse skills. When comparing the "skill initialization" to the "random initialization" in Section 4.2, we use the same model architecture for both methods. To pass skill $z$ to the Q function, value function, and policy, we simply concatenate $z$ to the current state $s_t$. As in  Haarnoja et al. (2018), epochs are 1000 episodes long. For all environments, episodes are at most 1000 steps long, but may be shorter. For example, the standard benchmark hopper environment terminates the episode once it falls over. Figures 2 and 5 show up to 1000 epochs, which corresponds to at most 1 million steps. We found that learning was most stable when we scaled the maximum entropy objective ($\mathcal{H}[A \mid S, Z]$ in Eq. 1) by $\alpha = 0.1$. We use this scaling for all experiments.

### C.1   Environments

Most of our experiments used the following, standard RL environments (Brockman et al., 2016): HalfCheetah-v1, Ant-v1, Hopper-v1, MountainCarContinuous-v0, and InvertedPendulum-v1. The simple 2D navigation task used in Figures 2a and 6 was constructed as follows. The agent starts in the center of the unit box. Observations $s \in [0, 1]^2$ are the agent's position. Actions $a \in [-0.1, 0.1]^2$ directly change the agent's position. If the agent takes an action to leave the box, it is projected to the closest point inside the box.

The cheetah hurdle environment is a modification of HalfCheetah-v1, where we added boxes with shape $H = 0.25m, W = 0.1m, D = 1.0m$, where the width dimension is along the same axis as the cheetah's forward movement. We placed the boxes ever 3 meters, start at $x = -1m$.

The ant navigation environment is a modification of Ant-v1. To improve stability, we follow Pong et al. (2018) and lower the gear ratio of all joints to 30. The goals are the corners of a square, centered at the origin, with side length of 4 meters: $[(2, 2), (2, -2), (-2, -2), (-2, 2), (2, 2)]$. The ant starts at the origin, and receives a reward of +1 when its center of mass is within 0.5 meters of the correct next goal. Each reward can only be received once, so the maximum possible reward is +5.

### C.2   Hierarchical RL Experiment

For the 2D navigation experiment shown in Figure 6, we first learned a set of skills on the point environment. Next, we introduced a reward function $r_g(s) = -\|s - g\|_2^2$ penalizing the distance from the agent's state to some goal, and applied the hierarchical algorithm above. In this task, the DIAYN skills provided sufficient coverage of the state space that the hierarchical policy only needed to take a single action (i.e., choose a single skill) to complete the task.

# D    MORE ANALYSIS OF DIAYN SKILLS

## D.1    TRAINING OBJECTIVES

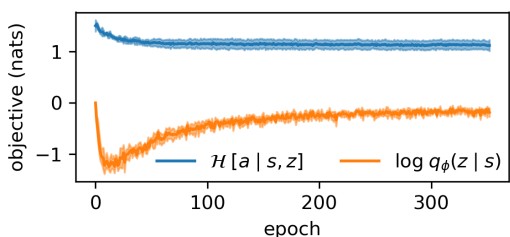

Figure 12: **Objectives**: We plot the two terms from our objective (Eq. 1) throughout training. While the entropy regularizer (blue) quickly plateaus, the discriminability term (orange) term continues to increase, indicating that our skills become increasingly diverse without collapsing to deterministic policies. This plot shows the mean and standard deviation across 5 seeds for learning 20 skills in half cheetah environment. Note that $\log_2(1/20) \approx -3$, setting a lower bound for $\log q_\phi(z \mid s)$.

To provide further intuition into our approach, Figure 12 plots the two terms in our objective throughout training. Our skills become increasingly diverse throughout training without converging to deterministic policies.

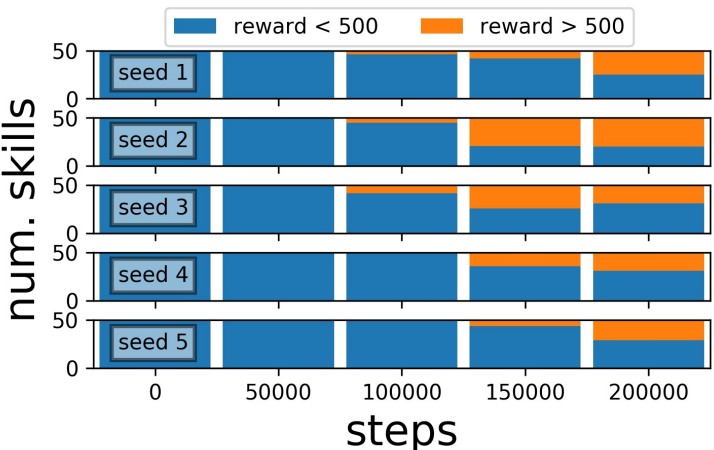

Figure 13: We repeated the experiment from Figure 2 with 5 random seeds to illustrate the robustness of our method to random seed.

To illustrate the stability of DIAYN to random seed, we repeated the experiment in Figure 2 for 5 random seeds. Figure 13 illustrates that the random seed has little effect on the training dynamics.

## D.2    EFFECT OF ENTROPY REGULARIZATION

**Question 9.** *Does entropy regularization lead to more diverse skills?*

To answer this question, we apply our method to a 2D point mass. The agent controls the orientation and forward velocity of the point, with is confined within a 2D box. We vary the entropy regularization $\alpha$, with larger values of $\alpha$ corresponding to policies with more stochastic actions. With small $\alpha$, we learn skills that move large distances in different directions but fail to explore large parts of the state space. Increasing $\alpha$

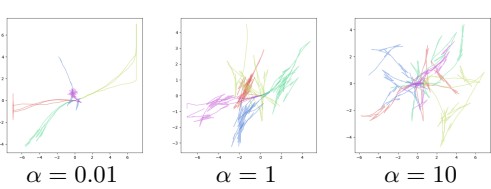

makes the skills visit a more diverse set of states, which may help with exploration in complex state spaces. It is difficult to discriminate skills when $\alpha$ is further increased.

## D.3 Distribution over Task Reward

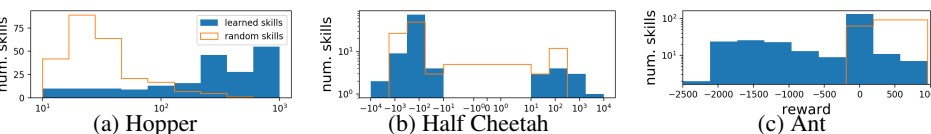

(a) Hopper    (b) Half Cheetah    (c) Ant

Figure 15: **Task reward of skills learned without reward**: While our skills are learned without the task reward function, we evaluate each with the task reward function for analysis. The wide range of rewards shows the diversity of the learned skills. In the hopper and half cheetah tasks, many skills achieve large task reward, despite not observing the task reward during training. As discussed in prior work (Henderson et al., 2017; Duan et al., 2016), standard model-free algorithms trained directly on the task reward converge to scores of 1000 - 3000 on hopper, 1000 - 5000 on cheetah, and 700 - 2000 on ant.

In Figure 15, we take the skills learned without any rewards, and evaluate each of them on the standard benchmark reward function. We compare to random (untrained) skills. The wide distribution over rewards is evidence that the skills learned are diverse. For hopper, some skills hop or stand for the entire episode, receiving a reward of at least 1000. Other skills aggressively hop forwards or dive backwards, and receive rewards between 100 and 1000. Other skills fall over immediately and receive rewards of less than 100. The benchmark half cheetah reward includes a control penalty for taking actions. Unlike random skills, learned skills rarely have task reward near zero, indicating that all take actions to become distinguishable. Skills that run in place, flop on their nose, or do backflips receive reward of -100. Skills that receive substantially smaller reward correspond to running quickly backwards, while skills that receive substantially larger reward correspond to running forward. Similarly, the benchmark ant task reward includes both a control penalty and a survival bonus, so random skills that do nothing receive a task reward near 1000. While no single learned skill learns to run directly forward and obtain a task reward greater than 1000, our learned skills run in different patterns to become discriminable, resulting in a lower task reward.

## D.4 Exploration

**Question 10.** *Does DIAYN explore effectively in complex environments?*

We apply DIAYN to three standard RL benchmark environments: half-cheetah, hopper, and ant. In all environments, we learn diverse locomotion primitives, as shown in Figure 3. Despite never receiving any reward, the half cheetah and hopper learn skills that move forward and achieve large task reward on the corresponding RL benchmarks, which all require them to move forward at a fast pace. Half cheetah and hopper also learn skills that move backwards, corresponding to receiving a task reward much smaller than what a random policy would receive. Unlike hopper and half cheetah, the ant is free to move in the XY plane. While it learns skills that move in different directions, most skills move in arcs rather than straight lines, meaning that we rarely learn a single skill that achieves large task reward on the typical task of running forward. In the appendix, we visualize the objective throughout training.

In Figure 16, we evaluate all skills on three reward functions: running (maximize X coordinate), jumping (maximize Z coordinate) and moving (maximize L2 distance from origin). For each skill, DIAYN learns some skills that achieve high reward. We compare to single policy trained with a pure exploration objective (VIME (Houthooft et al., 2016)). Whereas previous work (e.g., Pathak et al. (2017); Bellemare et al. (2016); Houthooft et al. (2016)) finds a single policy that explores well, DIAYN optimizes a *collection* of policies, which enables more diverse exploration.

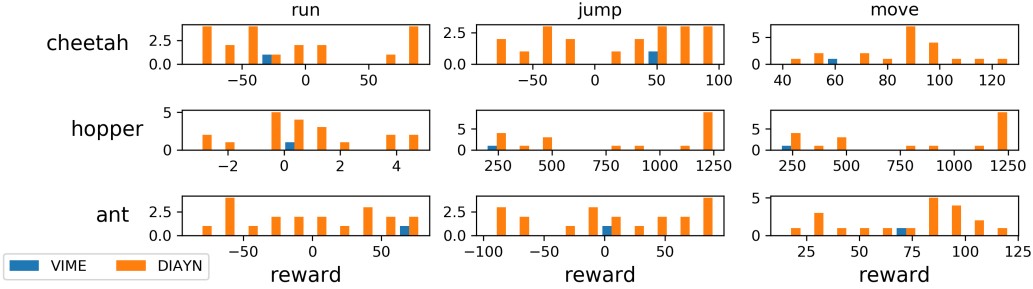

Figure 16: **Exploration**: We take DIAYN skills learned without a reward function, and evaluate on three natural reward functions: running, jumping, and moving away from the origin. For all tasks, DIAYN learns some skills that perform well. In contrast, a single policy that maximizes an exploration bonus (VIME) performs poorly on all tasks.

# E  LEARNING $p(z)$

We used our method as a starting point when comparing to VIC (Gregor et al., 2016) in Section 4.2. While $p(z)$ is fixed in our method, we implement VIC by learning $p(z)$. In this section, we describe how we learned $p(z)$, and show the effect of learning $p(z)$ rather than leaving it fixed.

## E.1  HOW TO LEARN $p(z)$

We choose $p(z)$ to optimize the following objective, where $p_z(s)$ is the distribution over states induced by skill $s$:

$$
\begin{aligned}
\mathcal{H}[S, Z] &= \mathcal{H}[Z] - \mathcal{H}[Z \mid S] \\
&= \sum_z -p(z) \log p(z) + \sum_z \mathbb{E}_{s \sim p_z(s)} \left[ \log p(z \mid s) \right] \\
&= \sum_z p(z) \left( \mathbb{E}_{s \sim p_z(s)} \left[ \log p(z \mid s) \right] - \log p(z) \right)
\end{aligned}
$$

For clarity, we define $p_z^t(s)$ as the distribution over states induced by skill $z$ at epoch $t$, and define $\ell_t(z)$ as an approximation of $\mathbb{E}[\log p(z \mid s)]$ using the policy and discriminator from epoch $t$:

$$
\ell_t(z) \triangleq \mathbb{E}_{s \sim p_z^t(s)}[\log q_t(z \mid s)]
$$

Noting that $p(z)$ is constrained to sum to 1, we can optimize this objective using the method of Lagrange multipliers. The corresponding Lagrangian is

$$
\mathcal{L}(p) = \sum_z p(z) \left( \ell_t(z) - \log p(z) \right) + \lambda \left( \sum_z p(z) - 1 \right)
$$

whose derivative is

$$
\begin{aligned}
\frac{\partial \mathcal{L}}{\partial p(z)} &= p(z) \left( \frac{-1}{p(z)} \right) + \ell_t(z) - \log p(z) + \lambda \\
&= \ell_t(z) - \log p(z) + \lambda - 1
\end{aligned}
$$

Setting the derivative equal to zero, we get

$$
\log p(z) = \ell_t(z) + \lambda - 1
$$

and finally arrive at

$$
p(z) \propto e^{\ell_t(z)}
$$

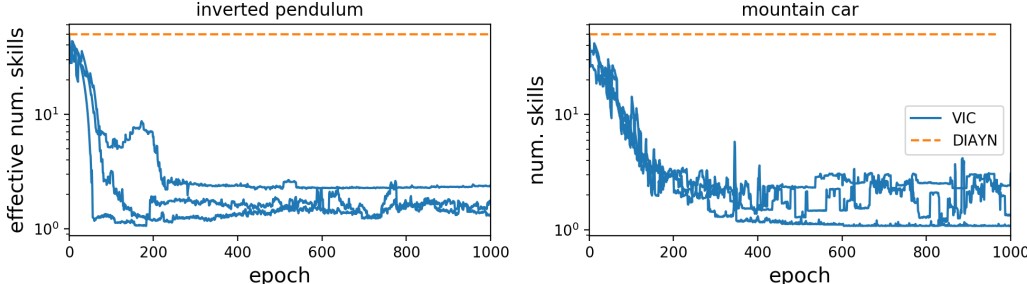

Figure 17: **Effect of learning** $p(z)$: We plot the effective number of skills that are sampled from the skill distribution $p(z)$ throughout training. Note how learning $p(z)$ greatly reduces the effective number on inverted pendulum and mountain car. We show results from 3 random seeds for each environment.

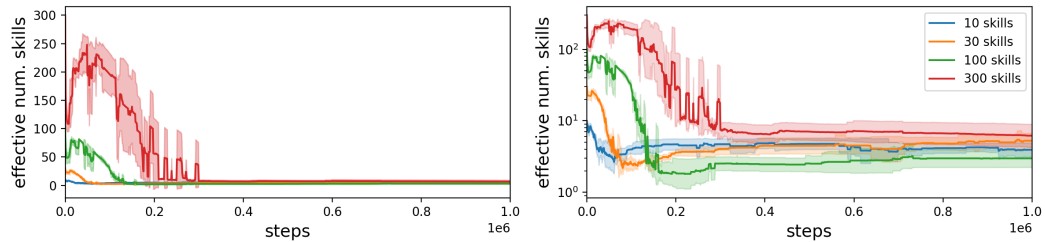

Figure 18: **Learning** $p(z)$ **with varying number of skills**: We repeat the experiment in Figure 4 for varying sizes of $z$. Regardless of the size of $z$, learning $p(z)$ causes the effective number of skills to drop to less than 10. The two subplots show the same data *(Left)* on a linear scale and *(Right)* logarithmic scale. We plot the mean and standard deviation across 3 random seeds.

## E.2    EFFECT OF LEARNING $p(z)$

In this section, we briefly discuss the effect of learning $p(z)$ rather than leaving it fixed. To study the effect of learning $p(z)$, we compared the entropy of $p(z)$ throughout training. When $p(z)$ is fixed, the entropy is a constant $(\log(50) \approx 3.9)$. To convert nats to a more interpretable quantity, we compute the effective number of skills by exponentiation the entropy:

$$\text{effective num. skills} \triangleq e^{\mathcal{H}[Z]}$$

Figure 17 shows the effective number of skills for half cheetah, inverted pendulum, and mountain car. Note how the effective number of skills drops by a factor of 10x when we learn $p(z)$. This observation supports our claim that learning $p(z)$ results in learning fewer diverse skills. Figure 18 is a repeat of the experiment in Figure 17, where we varying the dimension of $z$. Note that the dimension of $z$ equals the maximum number of skills that the agent could learn. We observe that the effective number of skills plummets throughout training, even when using a high-dimensional vector for $z$.

## F    VISUALIZING LEARNED SKILLS

### F.1    CLASSIC CONTROL TASKS

In this section, we visualize the skills learned for inverted pendulum and mountain car without a reward. Not only does our approach learn skills that solve the task without rewards, it learns multiple distinct skills for solving the task. Figure 19 shows the X position of the agent across time, within one episode. For inverted pendulum (Fig. 19a), we plot only skills that solve the task. Horizontal lines with different X coordinates correspond to skills balancing the pendulum at different positions along the track. The periodic lines correspond to skills that oscillate back and forth while balancing the pendulum. Note that skills that oscillate have different X positions, amplitudes, and periods. For

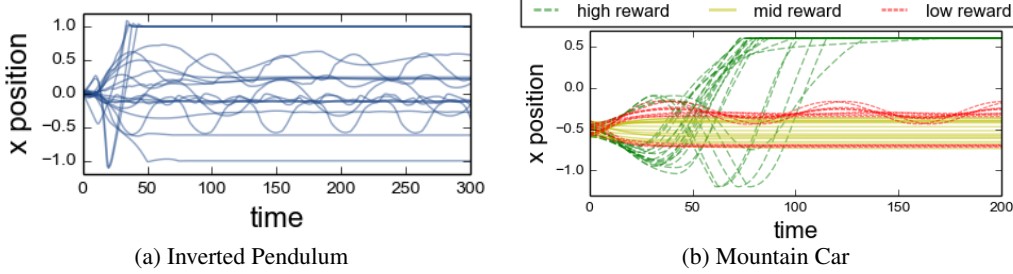

| (a) Inverted Pendulum | (b) Mountain Car |

Figure 19: **Visualizing Skills**: For every skill, we collect one trajectory and plot the agent's X coordinate across time. For inverted pendulum (top), we only plot skills that balance the pendulum. Note that among balancing skills, there is a wide diversity of balancing positions, control frequencies, and control magnitudes. For mountain car (bottom), we show skills that achieve larger reward (complete the task), skills with near-zero reward, and skills with very negative reward. Note that skills that solve the task (green) employ varying strategies.

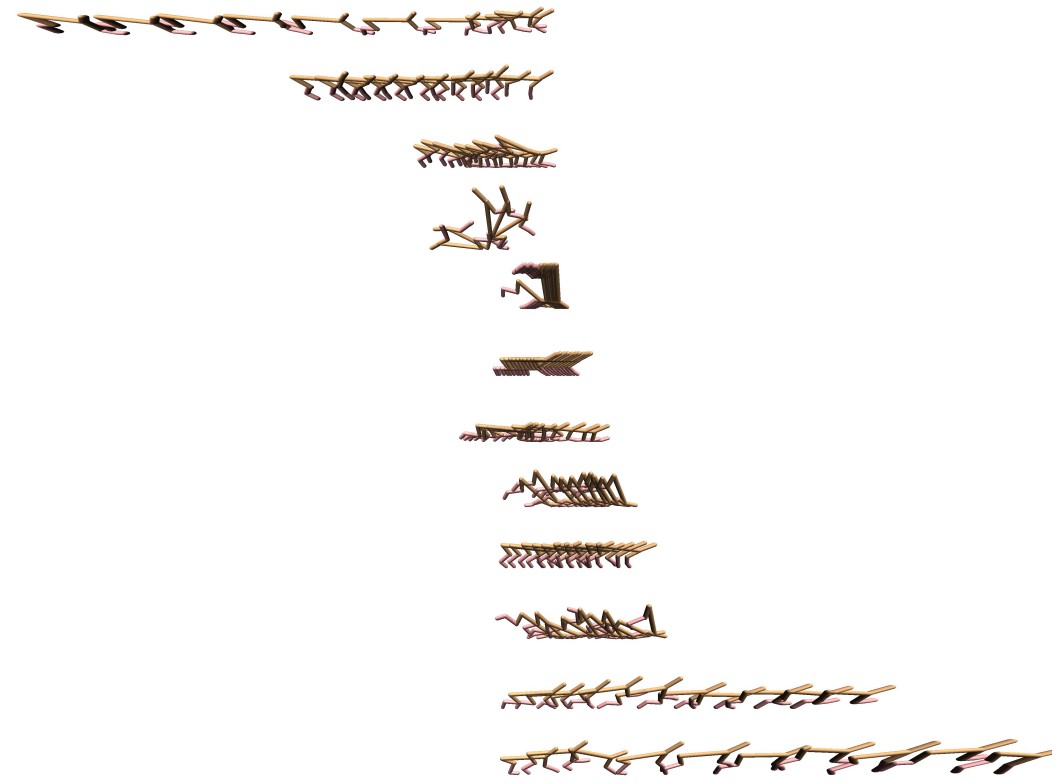

Figure 20: **Half cheetah skills**: We show skills learned by half-cheetah with no reward.

mountain car (Fig. 19b), skills that climb the mountain employ a variety of strategies for to do so. Most start moving backwards to gather enough speed to summit the mountain, while others start forwards, then go backwards, and then turn around to summit the mountain. Additionally, note that skills differ in when the turn around and in their velocity (slope of the green lines).

## F.2    SIMULATED ROBOT TASKS

Figures 20, 21, and 22 show more skills learned *without reward*.

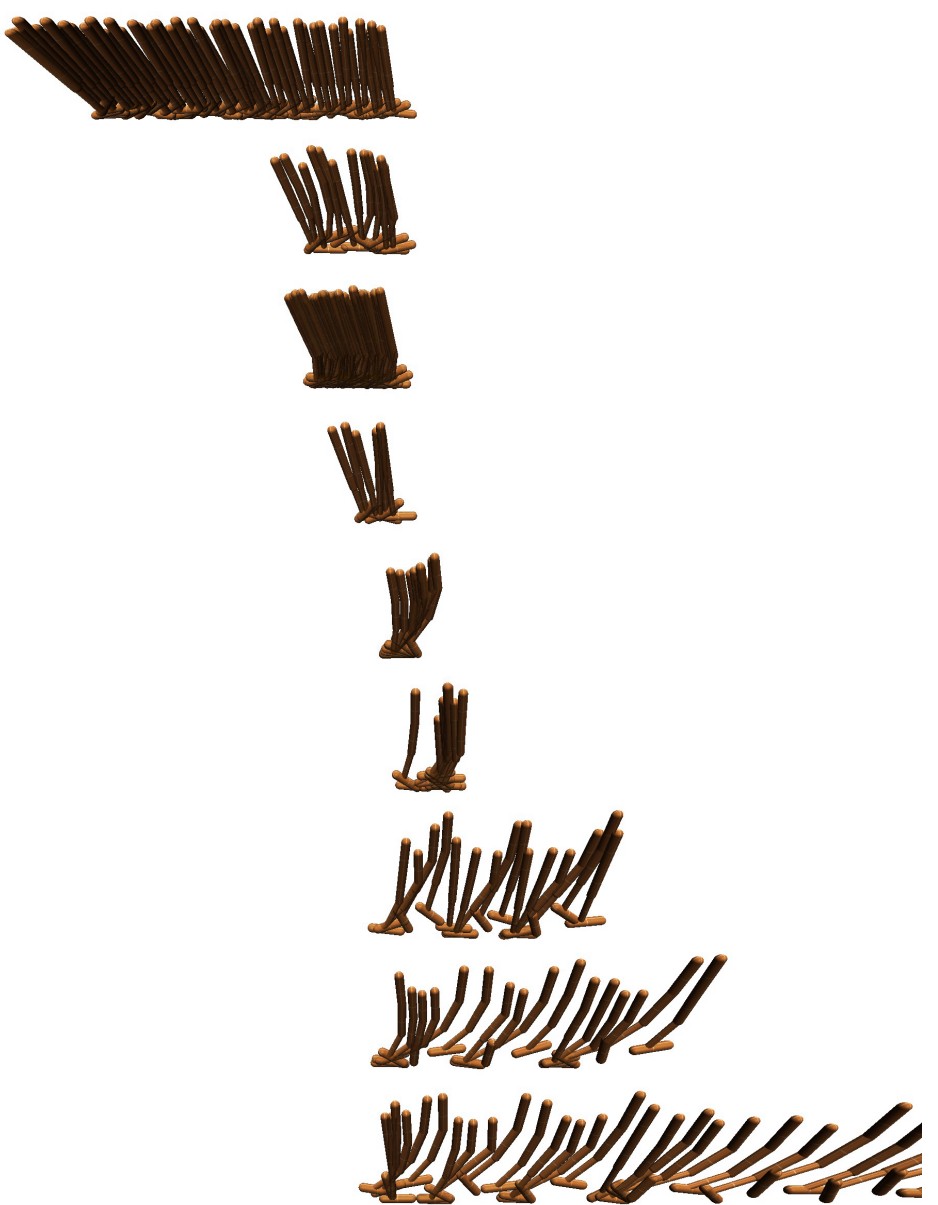

Figure 21: **Hopper Skills**: We show skills learned by hopper with no reward.

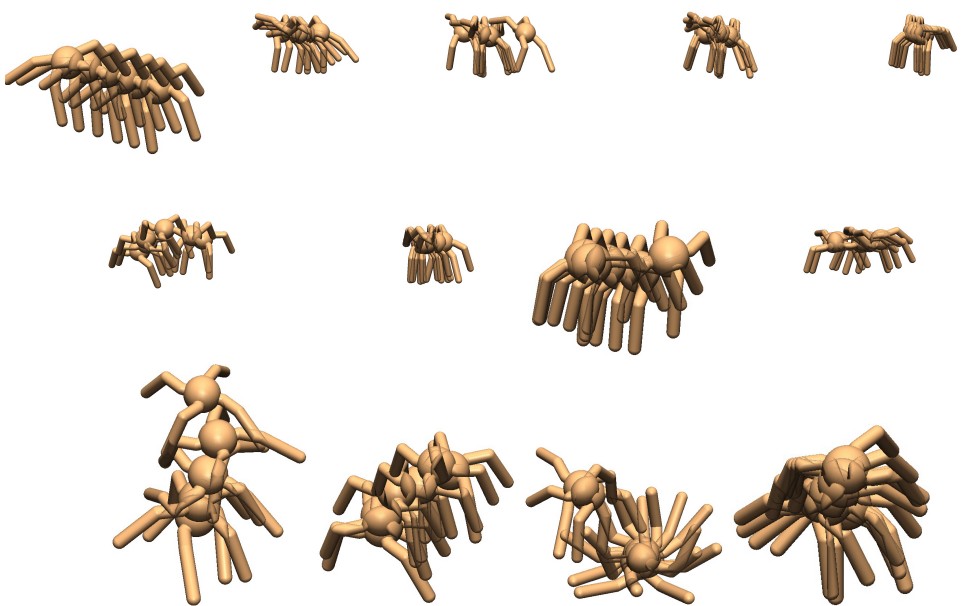

Figure 22: **Ant skills**: We show skills the ant learns without any supervision. Ant learns *(top row)* to move right, *(middle row)* to move left, *(bottom row, left to right)* to move up, to move down, to flip on its back, and to rotate in place.

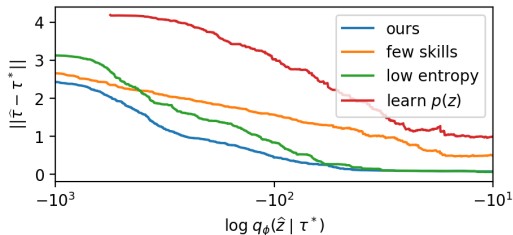

Figure 23: **Imitating an expert**: Across 600 imitation tasks, we find our method more closely matches the expert than all baselines.

## G   IMITATION LEARNING

Given the expert trajectory, we use our learned discriminator to estimate which skill was most likely to have generated the trajectory:

$$\hat{z} = \arg\max_{z} \Pi_{s_t \in \tau^*} q_\phi(z \mid s_t)$$

As motivation for this optimization problem, note that each skill induces a distribution over states, $p^z \triangleq p(s \mid z)$. We use $p^*$ to denote the distribution over states for the expert policy. With a fixed prior distribution $p(z)$ and a perfect discriminator $q_\phi(z \mid s) = p(z \mid s)$, we have $p(s \mid z) \propto q_\phi(z \mid s)$ as a function of $z$. Thus, Equation G is an M-projection of the expert distribution over states onto the family of distributions over states, $\mathcal{P} = \{p^z\}$:

$$\arg\min_{p^z \in \mathcal{P}} D(p^* \mid\mid p^z) \tag{5}$$

For clarity, we omit a constant that depends only on $p^*$. Note that the use of an M-projection, rather than an I-projection, helps guarantee that the retrieved skill will visit all states that the expert visits (Bishop, 2016). In our experiments, we solve Equation 5 by simply iterating over skills.

### G.1   IMITATION LEARNING EXPERIMENTS

The "expert" trajectories are actually generated synthetically in these experiments, by running a different random seed of our algorithm. A different seed is used to ensure that the trajectories are not actually produced by any of the currently available skills. Of course, in practice, the expert trajectories might be provided by any other means, including a human. For each expert trajectory, we retrieve the closest DIAYN skill $\hat{z}$ using Equation 4.2.3. Evaluating $q_\phi(\hat{z} \mid \tau^*)$ gives us an estimate of the probability that the imitation will match the expert (e.g., for a safety critical setting). This quantity is useful for predicting how accurately our method will imitate an expert before executing the imitation policy. In a safety critical setting, a user may avoid attempting tasks where this score is low. We compare our method to three baselines. The "low entropy" baseline is a variant on our method with lower entropy regularization. The "learned $p(z)$" baseline learns the distribution over skills. Note that Variational Intrinsic Control (Gregor et al., 2016) is a combination of the "low entropy" baseline and the "learned $p(z)$" baseline. Finally, the "few skills" baseline learns only 5 skills, whereas all other methods learn 50. Figure 23 shows the results aggregated across 600 imitation tasks. The X-axis shows the discriminator score, our estimate for how well the imitation policy will match the expert. The Y-axis shows the true distance between the trajectories, as measured by L2 distance in state space. For all methods, the distance between the expert and the imitation decreases as the discriminator's score increases, indicating that the discriminator's score is a good predictor of task performance. Our method consistently achieves the lowest trajectory distance among all methods. The "low entropy" baseline is slightly worse, motivating our decision to learn maximum entropy skills. When imitating tasks using the "few skills" baseline, the imitation trajectories are even further from the expert trajectory. This is expected – by learning more skills, we obtain a better "coverage" over the space of skills. A "learn $p(z)$" baseline that learns the distribution over skills also performs poorly. Recalling that Gregor et al. (2016) is a combination of the "low entropy" baseline and the "learn $p(z)$" baseline, this plot provides evidence that using maximum entropy policies and fixing the distribution for $p(z)$ are two factors that enabled our method to scale to more complex tasks.

