# OpenReview forum: "Diversity is All You Need: Learning Skills without a Reward Function"
_ICLR.cc/2019/Conference_

### Official Review · AnonReviewer3 · 2018-11-03
**Strong paper with interesting contributions, but with slightly over-stated claims**

**Rating:** 7
**Confidence:** 4

**Review:**

This paper proposes a method for learning skills in absence of a reward function. These skills are learned so that the diversity of the trajectories produced by each skill is maximised. This is achieved by having a discriminator attempting to tell these skills apart. The agent is rewarded for visiting states that are easy to distinguish and the discriminator is trained to better infer the skills from states visited by the agent. Furthermore, a maximum entropy policy is used to force the skills to be diverse. The proposed method is general and any RL algorithm with entropy maximisation in the objective can be used, the implementation in the paper uses the Soft Actor Critic method.

The problem that they are tackling is interesting and is of clear value for obtaining more generalisable RL algorithms. The paper is overall clear and easy to follow, the results are interesting and potentially useful, although I have some reservations regarding how they assess this usefulness in the current version of this paper.
Structure-wise, I would say that the choice of writing the paper in the form of a Q&A, with very brief explanations and details was more distracting and at times unnecessary than I liked (e.g. Question 7 could move to Appendix as it is quite trivial).

I really appreciated how much care has been taken to discuss differences with the closest prior work, Variational Intrinsic Control (VIC) by Gregor et al.
One such difference is that their prior distribution over skills is not learnt. While there are good arguments by the authors about why this is appealing (e.g. it prevents collapsing to sampling only a few skills), I feel this could be also quite a limitation of their method. This assumes that you have a good a-priori knowledge and assumptions regarding how many skills are useful or needed in the environment. This is unlikely to be the case in complex environments, where you first need to learn simple skills in order to explore the environment, and later learn to form new more complex skills. During this process, you might want to prune simplistic skills after you learnt more abstract and complex ones, for instance in the context of continual learning. I understand this could be investigated in future work, but I feel they take a rather optimistic take on this problem.

Overall, the use case for the proposed method is slightly unclear to me. While the paper claims to allow diverse set of skills to be learnt, it is highly dependent on learning varied action sequences that help you visit different part of state space, regardless of their usefulness. This means there could be learn a lot of skills that capture part of the state space that is not useful or desirable for downstream tasks. While there is a case made for DIAYN being a stepping stone for imitation learning and hierarchical RL, I don’t find the reported experiments for imitation learning and HRL convincing. In the imitation learning experiment, the distance (KL divergence) between all skills and the expert data is computed and the closest skill is then chosen as the policy imitating the expert. The results are weak and no comparisons with any LfD baselines are reported. The HRL experiments also lack comparisons to any other HRL baseline. I feel that this section is rather weak, especially compared to the rest of the paper, and I am not sure it achieves much.

As a general comment, the choice of reporting the training progress using “hours spent training” is an peculiar choice which is never discussed. I understand that for methods with varying computational costs this might be a fairer comparison but it would be perhaps good to also report progress against number of required environment interactions (including pre-training).
Another assumption made is that the method is valuable in situations where the reward function is expensive to compute and the unsupervised pre-training is free (somewhat easing the large amount of pre-training required). However, it would have been interesting to see examples of such environments in their experiments supporting these claims, as this assumption is not valid for the chosen MuJoCo environments.

Despite these comments, I still feel this is valuable work, that can clearly inspire further relevant work and deserves to be presented at ICLR.
It presents a solid contribution, given its technical novelty, proposed applications and its overall generality.
However, the paper could use more convincing experiments to support its claims.

Additional comments and typos:
- Figure 5 lack error bars across the 5 random seeds and are crucial to assess whether this performance difference is indeed significant given the amount of pre-training required.
- Figure 7’s title and caption is missing...
- typo: page 3, last paragraph “...mutual information between skills and states, **I(S; Z )**” not I(A; Z)
- typo: page 7 paragraph next to Figure 6 “...whereas DIAYN explicitly **learns** skills that effectively partition the state space”
- typo: page 7 above Figure 8 “...make them **exceedingly** difficult for non- hierarchical RL algorithms.”

---

> ### Author Response · Authors · 2018-11-19
> **Author response**
>
> Thanks for all the feedback!
>
> Learning p(z): We would like that emphasize that prior work (VIC [Gregor et al]) also requires the user to choose the number of skills. Choosing this parameter is analogous to choosing K in K-means. While we can propose various heuristics, the right choice ultimately depends on the problem at hand. Empirically, we found that a 50-dimensional categorical distribution worked well in all environments we tested. Your question hints at a deeper question: can we recursively or hierarchically compose skills to learn behaviors of increasing complexity? While this is beyond the scope of this work, we think it is an exciting direction for future research, and encourage work in this direction.
>
> "Useless skills": We agree that without explicitly biasing our skills towards certain types of behaviors, we are likely to end up with many skills that are not useful for a particular given task. We presented one way to bias skill discovery in Section 4.2.2 (Question 7), and showed experimentally that it enabled better performance on hierarchical tasks (Figure 8 (right), purple line). We should emphasize that including a task-specific biases skill may cause poor generalization across tasks.
>
> LfD and HRL baselines: We will run additional experiments comparing to imitation learning and hierarchical RL baselines in the final version.
>
> For Figure 8, we plotted over time instead of iterations because TRPO and SAC have very different costs per iteration (TRPO is substantially most costly). We will include a plot over time in the final paper, but caution that TRPO will be run for many fewer iterations than SAC. We'll also fix the typos and clarifications you've suggested.

---

### Official Review · AnonReviewer2 · 2018-11-04
**Review - Diversity is All You Need: Learning Skills without a Reward Function**

**Rating:** 7
**Confidence:** 3

**Review:**

*Pros:*
-	Mostly clear and well-written paper
-	Technical contribution (learning many diverse skills) is principled and well-justified (although the basic idea builds on a large body of prior work from related fields as also evidenced from the reference list)
-	Extensive emperical evaluation on multiple tasks and different scenarios (mainly toy examples) showing promising results.

*Cons:*
-	The main paper assumes detailed knowledge of the actor critic setup to fully follow and appreciate the paper (a few details provided in the appendix)
-	p(z): it is not entirely clear to me how the dimensionality of z should be chosen in a principled manner aside from brute force evaluation (as in 4.2.2; which does not go beyond a few hundreds). What happens for many skills and would learning p(z) be preferable in this scenario?
-	Note: The work has been in the public domain for some time thereby limiting the apparent novelty. This has not influenced my decision as per ICLR policy.

*Significance*: I think this work would be of interest to the ICLR crowd despite it having been in the public domain for a some time. It provides a simple objective for training RL models in an unsupervised manner by learning multiple diverse skills and contributes with an extensive and convincing empirical evaluation which will surely have a lasting impact in the RL subfield.

*Further comments/questions*:
-	The authors assume that only states and not actions are observable. Intuitively, it would seem easier to obtain the desired results if the actions are also available. Could the authors perhaps clarify why it is reasonable to assume that the actions are not observable to the planner when evaluating the objective in Eq 1?  Similarly, I’d like some insight into the behaviour of the proposed method if actions are also available (and how it differs from prior art in this case)?
-	I’d suggest enforcing consistently in the way variation across random seeds is visualised in the figures (e.g. traces in fig 4, no indication in fig5, shaped background in fig 6).
-	I’d suggest making it explicit what $\theta$ refers to in Eq. 1 (and provide some details about the SAC setup for completeness, as previously mentioned)
-	Minor typos etc: {p2, l6} missing word, “SAC” never defined, “DOF” never defined, and a few other typos/punctuation issues throughout.

---

> ### Author Response · Authors · 2018-11-19
> **Author response**
>
> Thanks for your helpful comments!
>
> Dimension of z: The dimension of the skill indicator, z, is a hyperparameter. If the dimension is too small, then you cannot learn many skills. If the dimension is too large, many skills are not distinguishable. While we found that a 50 dimensional categorical distribution worked well for all environments we tested, the right choice ultimately depends on the problem at hand.
>
> Learning p(z): To answer your questions about learning large numbers of skills, we repeated the experiment in Figure 4, varying the number of skills from 10 to 300. Results are now included in Appendix E, Figure 18. We found increasing the number of skills does not solve the "rich get richer" problem. Even when learning 300 skills, the effective number of distinguishable skills quickly drops to be less than 10.
>
> Why not condition on actions? We agree that it would be "easier" to maximize our mutual information objective if we conditioned the discriminator on both states and actions, rather than just states. In fact, the data processing inequality tells us this: the mutual information between (state, action) pairs and latents I(S, A; Z) is an upper bound for the mutual information between states and latents I(S; Z). However, the ease of taking discriminable actions is precisely the problem with this objective. In preliminary experiments, we found that conditioning the discriminator on both states and actions substantially increased the number skills that were distinguishable in action space but not state space, effectively maximizing I(A; Z). This result is problematic because, for skills to be useful for downstream tasks, they must consistently change the state.
>
> We have fixed the typos and reworded the method section to be more accessible to readers unfamiliar with SAC.

---

### Official Review · AnonReviewer1 · 2018-11-06
**Clearly-written, well-demonstrated learning and application of unsupervised skills via information-theoretic / entropic methods**

**Rating:** 8
**Confidence:** 4

**Review:**

The authors propose a learning scheme for the unsupervised acquisition of skills. These skills are then applied to (1) accelerate reinforcement learning to maximize a reward, (2) perform hierarchical RL, and (3) imitate an expert trajectory.

The unsupervised learning of skills maximizes an information theoretic objective function. The authors condition their policy on latent variable Z, and the term skill refers to a policy conditioned on a fixed Z. The mutual information between states and skills is maximized to ensure that skills control the states, while the mutual information between actions and skills given the state is minimized, to ensure that states, not actions distinguish skills. The entropy of the mixture of policies is also maximized. Further manipulations on this objective function enable the scheme to be implemented using a soft actor-critic maximizing a pseudo reward involving a learned skill discriminator.

The authors clearly position their work in relation to others, and especially point out the differences to the most similar work, namely Gregor et al 2016. These differences while seemingly minor end up providing exceptional improvement in the number of skills learned and the domains tackled.

The question-answer style is somewhat unconventional. While the content comes across clearly, the flow / narrative is a bit broken.

Overall, I believe that applicability of the work is very wide, touching inverse RL, hierarchical RL, imitation learning, and more. The simulational comparisons are also very useful.

However, there is an issue that I'd like to see addressed:
Fig 8: In a high-dimensional task, namely 111D ant navigation, DIAYN performs slightly worse than others. Incorporating a prior on useful skills makes DIAYN perform much better. Here, apart from the comparision with other state of the art RL methods, the authors should also compare to VIC. Indeed one of the key differences to VIC was the uniform prior on skills, which the authors now break albeit in a slightly different way. Thus, it is essential to also show the performance of VIC, and comment on any similarities / differences. The relation of this prior to the VIC prior should also be made clear. Further, the performance of VIC on the half cheetah hurdle should be also be shown.

If the above issue is addressed, I strongly recommend that the work be presented at ICLR.

Minor issues / typos:
pg 1: "policy that alters that state of the environment" to "policy that alters the state of the environment"
pg 3: "mutual information between skills and states, I(A; Z)" to "mutual information between skills and states, I(S; Z)"
pg 4: "guaranteeing that is has maximum entropy" to "guaranteeing that it has maximum entropy"
pg 4: " soft actor critic" to " soft actor critic (SAC)" since SAC is used later.
pg 5: full form of VIME not introduced
Fig 5: would be good to also show the variance as a shaded area around the mean.
pg 7: "whereas DIAYN explicitly skills that effectively partition the state space" ?

---

> ### Author Response · Authors · 2018-11-19
> **Author response**
>
> Thanks for your comments! We are currently running the requested comparison to VIC in the hierarchical RL experiments, and will update the paper once completed. We have also corrected the typo that you mentioned.

---

### Meta-Review · Area_Chair1 · 2018-12-14
**Interesting and well-executed paper**

**Confidence:** 4
**Recommendation:** Accept (Poster)

**Metareview:**

There is consensus among the reviewer that this is a good paper. It is a bit incremental compared to Gregor et al 2016. This paper show quite better empirical results.